# Use of Reaction Force to Evaluate Older Adults’ Gait Patterns While Using a Walker to Walk

**DOI:** 10.3390/geriatrics4030043

**Published:** 2019-07-14

**Authors:** Po-Chan Yeh

**Affiliations:** Institute of Creative Design and Management, National Taipei University of Business, No. 100, Sec. 1, Fulong Rd., Pingzhen Dist., Taoyuan City 324, Taiwan; hilester@ntub.edu.tw

**Keywords:** gait pattern, older adults, walker

## Abstract

Walking is the most common activity in daily life. As people age, however, they begin to become imbalanced and need the assistance of mobility devices for walking, such as walkers. However, clinical gait measurement requires a lot of equipment to be worn; as walker users are seniors or disabled, this may cause them to be troubled in the assessment. Thus, this study used four load cells on the walker to estimate gait status. To understand the difference between the three groups of the Berg Balance Scale (BBS), 60 volunteers, who served as the subjects, were divided into three groups according to BBS scores, 20 volunteers for each group. Data were obtained from four load cells; walker users were divided per the BBS to observe their stance, swing phases, and support force while walking. The results of the study found that participants in the study were able to walk smoothly with the walker, and differences between the three groups in stance, swing phases, and support force were observed. The main findings of this study were: (1) While walking, the stance and swing phases could be stabilized by the evaluated gait; and (2) even if the user can walk stably, body function can be evaluated by the support force. We hope that our method will be widely applied in the design of mobility devices and in the evaluation of seniors’ care; we also hope our study will contribute to increasing knowledge, generally, in this field.

## 1. Introduction

Walking is the most common activity in daily life [1]. The movement of one leg from one heel-strike to the next is called a gait cycle (GC); each GC has stance and swing phases [2]. The stance phase is the phase from heel-strike to toe-off, and the swing phase is that from toe-off to the next heel-strike by the same foot. In a stable gait pattern, the GC parameters of both legs are symmetrical and occur at a stable speed [3]. To achieve the goal of walking independently, gait control requires four conditions: (1) continuously moving the center of mass (COM) smoothly and stably out of the base of support; (2) building a new base of support during a new GC; (3) pushing off the body while maintaining balance; and (4) adopting the goal of walking and meeting environmental requirements [4].

As the body ages, pathological changes in the brain and experiences of falling cause a decline in the ability to balance [5,6,7]. Several studies have noted that senior individuals in relatively poor health and people with lower limb injuries are more likely to be unstable when walking [8,9]; a loss of balance is the main cause of falls among older adults [10,11]. Independent mobility is one of the most important factors affecting the quality of life of seniors and other clinical populations who need assistive devices such as walkers [12,13]. According to Kay and Hugh [14], the majority of walker users are older adults aged 60 years and older. Users must lift the walker with their hands and use it to support their body as they propel themselves forward.

A walking motion with a walker is a repeated motion, just as walking normally is [15]. The process of using a walker is as follows: the user first holds onto the walker with both hands and moves the walker half a step forward; the user then moves the injured leg forward, maintaining his/her center of gravity above the center of the walker; and, finally, the user moves the healthy leg forward. This process is repeated when walking [16]. The Berg Balance Scale (BBS) is the most commonly used tool for evaluating the balance of patients or older adults [17]. BBS scores of 0–20, 21–40, and 41–56 indicate high, medium, and low fall risks, respectively [17]. Many studies have proven that high-risk groups are more likely to fall and become hospitalized [18], and that walking speed is an extremely important factor in determining balance [19].

Studies on gait and balance mainly fall into two categories, namely, clinical gait measures and instrumental gait analysis. Clinical gait measures generally determine walking speed, which can be defined as the amount of time it takes subjects to walk a specific distance [20,21], or as the distance that subjects walk within a specific amount of time [22,23]. Instrumental gait analysis uses a proportionality scheme applied on the combination of pressure and force plate data [24,25,26]. In gait analysis, Mohammed et al. [20] collected pressure and force measurements during the gait cycle, while Leung and Yeh [27] used strain gauges or load cells installed on the four legs of a walker to measure the pattern and magnitude of forces. Furthermore, according to Alwan et al. [28], the process of placing a walker can be used to estimate the time from heel-strike to toe-off, and even to estimate the entire GC.

According to the above literature, the study of gait requires the participant to wear a number of devices. However, most walker users are older adults or disabled, and the drawback of these measurements is that they are obtained with specialized laboratory equipment, and wearing a lot of measuring equipment may cause older adults or disabled people trouble [29,30]. Additionally, the results would also be expected to present significant differences between observing the users under natural environment and laboratory conditions. Compared with field experiments and laboratory experiments, the advantage of field experiments is that several experimental variables can be observed simultaneously [31]. Nishdia et al. [32] considered that monitoring the walking of elderly people under a natural-environment-based setting is better than using laboratory settings. Even though BBS scores are used to evaluate the balance of patients or older adults, floor and ceiling effects more easily occur in low-risk and high-risk groups [33].

In order to reduce the requirement to wear a large number of instruments, this paper proposes a method of evaluation to analyze groups of walker users, divided as per the BBS, to understand differences in their stance and swing phases and support forces. The abovementioned theoretical foundation of gait is applied [2,20,21,28] using measurement instruments based on Deschamps et al. [24], Mohammed et al. [20], Saraswat et al. [25], Leung and Yeh [27], and Bruening, et al. [13] to observe the patterns and magnitude of forces when walker users shift their body weight to the walker during the stance and swing phases, as well as to observe differences in support forces. In order to estimate gait status, the experimental tests in this study were evaluated in the following way:Decomposing the gait into stance, swing phases, and support force by reaction;Whether there are significant differences between two steps in stance, swing phases, and support force while using a walker;Whether there are significant differences between two steps in stance, swing phases, and support force between BBS groups while using a walker.

## 2. Methods

### 2.1. Participants

The experimental unit was notified of the requirements and purpose of this study before the experiment. The school sent an official document to the experimental unit, which requested that it assist with recruiting suitable volunteers. According to Gay, Mills and Airasian [34], a minimum of 30 subjects are required for comparative and relationship studies. Sixty volunteers, who served as the subjects, were divided into three groups according to BBS scores, with 20 volunteers for each group. Users included 60 experienced participants aged 65–95 years (mean = 82.3; BBS scores: high risk = 17.20, *SD* = 2.86; medium risk = 33.95, *SD* = 5.17; low risk = 50.79, *SD* = 5.12) who reported frequently using a walker independently and who had over 3 months of experience of use. The 60 older adults were enrolled in two senior centers in Taipei City. Informed consent was obtained from all subjects after a full explanation of this study, and its procedures were provided to them. All subjects participated anonymously.

### 2.2. Experimental Walker

According to Leung et al. [27], who installed load cells (JIHSENSE S100) on the walker and evaluated gait by the theory of reaction force according to Mohammed et al. (2016), an experimental walker was designed for this study. The adjustable-height (72–82 cm) experimental walker was aluminum (width, 52 cm; weight, 4 kg), and the data of the load cells installed on each of the walker’s four feet were transmitted to the computer. The experimental walker was used for the experimental tasks, and load cells were used to convert force into electrical signals. The software then “zeroed” for each strain gauge while the walker was unloaded.

### 2.3. Task Specification

The collection sessions each lasted about 40 min for each subject. The experimental data were collected using the measurement devices from the experimental walker and load cells. 

Before the experiment, the experiment unit provided subjects’ BBS scores, and a researcher measured the weight, height, and larger trochanter height of each subject. After that, the researcher adjusted the walker height according to each subject’s trochanter height. At the beginning of the experiment, the subjects were using walkers at their usual walking speed individually in a quiet room at a temperature of approximately 25 °C. When subjects were walking stably, the researcher recorded data for 3 m walking segments—the layout of the experimental setting is shown in Figure 1—and, for safety reasons, a registered nurse was present for all trials. A computer was used to simultaneously record vertical force data from load cells at 1000 Hz (see Figure 2).

### 2.4. Data Analysis

The study data mainly comprised the GC of the subjects when using a walker to walk. The stance and swing phases when using the walker were defined as follows. The stance phase is based on the theory of Peat et al. [3] and Aikaterini et al. [15], and was defined as from the moment one leg of the walker is lifted from the ground until the walker is stably placed on the ground. The swing phase is the phase when the body is moving. According to Mohammed et al. [20] and Aikaterini et al. [15], the swing phase starts after the point when the walker is placed stably on the ground to when one leg of the walker is lifted from the ground. One stance phase and one swing phase form one GC. This study used the time required for two GCs to evaluate the pattern and magnitude of forces.

To assess the process of walking, we examined three types of data: (1) the maximum supporting force (N) of two steps as the subjects walked using the walker; (2) the steady state of two steps (the amount of time from the point when the walker was placed stably on the ground to the point when the walker was lifted); and (3) the swing phase of the two steps (the time from the point when one leg of the walker was placed on the ground to the point when all four legs of the walker were on the ground).

Analysis was conducted using the SPSS-PC statistical software package. First, descriptive statistics were computed. Next, the multivariate analysis of variance (MANOVA) was used to determine whether there were significant differences (*p* < 0.05) in the support forces, stance phase time, and swing phase between the two steps and BBS groups while using a walker to walk. If the MANOVA showed a significant result, the least significant difference (LSD) method was used to conduct a post hoc test. Finally, Pearson’s product–moment correlation was adopted to examine the correlations among the stance phase, swing phase, support forces, and Berg group. The results were considered statistically significant if *p* < 0.05.

## 3. Results

If, after conducting a multivariate analysis of variance (MANOVA), there was no significant difference in stance phases, swing phases, and support force between the two steps while using a walker to walk, it can be inferred that the subject operates the walker in a stable gait (see Table 1).

There was a significant difference in the groups divided based on the BBS with respect to the stance phase (F = 123.78, *p* = 0.00 < 0.05) and swing phase (F = 54.41, *p* = 0.00 < 0.05). The post hoc analysis found that the high-risk group (M = 3.54, SD = 0.51) and medium-risk group (M = 3.73, SD = 0.81) spent shorter amounts of time in the stance phase than the low-risk group (M = 1.67, SD = 0.53); compared with the high-risk group (M = 1.28, SD = 0.42) and low-risk group (M = 2.08, SD = 0.72), the medium-risk group (M = 2.64, SD = 0.60) spent more time in the swing phase. 

There was also a significant difference in support force (F = 8.52, *p* = 0.00 < 0.05). The post hoc analysis divided participants into three groups. The high-risk group required the greatest support forces (M = 189.14, SD = 67.89), followed by the medium-risk group (M = 127.62, SD = 43.96) and then the low risk group (M = 101.34, SD = 61.41). This indicates that a lower BBS score indicates the requirement for more support force from the walker (see Table 1).

There was a significant correlation (r = −0.67, *p* < 0.01) between the Berg group and stance phases, indicating that a good physical condition was associated with a shorter stance phase time. Moreover, there was also a significant correlation (r = −0.52, *p* < 0.01) between the Berg group and support forces, indicating that good physical condition was associated with lower support forces. Moreover, the correlation (r = 0.41, *p* < 0.01) between the swing phase and Berg group was statistically significant and positive, indicating that a good physical condition was associated with the swing phase time.

## 4. Discussion

This study used the BBS as a basis for dividing users into groups, and used load cells to determine support force and the time spent in the stance and swing phases, evaluating the difference between the walking time and support force of the three groups and using this analysis to reduce gait assessment while wearing too many devices.

The subjects sought in this study were divided into three groups by BBS. All three groups of older adults were able to walk on their own. The results showed that there was no difference between the stance and swing phases in the two steps, according to Peat et al. [3]. According to gait theory, in a stable gait pattern, the GC parameters of both legs are symmetrical and occur at a stable speed, and, from the results, it can be inferred that the subjects operated their walkers in a stable gait.

According to the results of the BBS groups, the high- (M = 3.54) and medium-risk (M = 3.73) spent more time to move the walker; it is assumed that they had a worse physical condition, and so they increased the period of their stance. During swing phase, it is assumed that high risk group (1.28) had worse physical condition, so they shortened the period of swing. It is noteworthy that the medium-risk group used the walker to support their body, therefore the medium-risk group (M = 2.64) could spend more time than the low-risk group in the swing phase. However, no matter the walk speed, in terms of support force, the high-risk group needed more support force from the walker when walking than the medium- and low-risk groups. There was a significant difference in support force of BBS groups; in this study, it was found that the medium-risk group had a lower swing state than the low-risk group, which, according to Burnfield et al. [8], Hausdorff et al. [9] has the potential to increase the risk of falls. The specific results and recommendations of this study are as follows:Stabilization of the gait by stance and swing phases of the walking aid while walking;Even if the user can walk stably, the body function can be evaluated by the support force.

Stable walking is a daily behavior, but it is a very important issue for older adults and weak individuals. Most gait tests are carried out in the laboratory and require the participant to wear a large number of instruments, and older adults may be rejected when conducting a test evaluation because of differences of self-esteem and cognition. However, this was not found in the early stage of degeneration of bodily functions, resulting in subsequent injuries or even medical costs. The evaluation of gait in this study can indeed reduce the requirement to wear a large number of instruments. Meanwhile, the gaits of users while using walkers could also be predicted. If older adults can be initially evaluated by a caregiver at a nursing center/rehabilitation center, the caregiver could monitor the users’ gaits at any time. For instance, this study discovered that the medium-risk group needs special concern when they are acting alone. Moreover, the data will be much more accurate if they are obtained from the assessment of actual walking gaits under natural environment conditions instead of the laboratory. This is an exciting first step, but daily walking includes hills, stairs, and even ground materials and other factors, all of which affect stability when walking. Thus, future research is obviously required. We hope that our method will be widely applied in the design of mobility devices and in evaluation of the care of older adults. We also hope that our study will contribute to increased knowledge in this field.

## Figures and Tables

**Figure 1 geriatrics-04-00043-f001:**
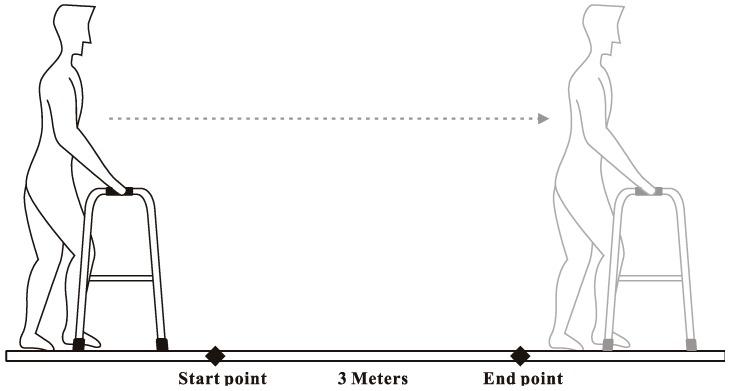
The layout of the experimental setting.

**Figure 2 geriatrics-04-00043-f002:**
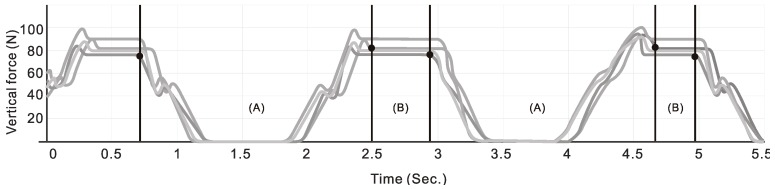
Stance phase (**A**) and swing phases (**B**) for each step.

**Table 1 geriatrics-04-00043-t001:** Analysis of variance (ANOVA) table for two steps and Berg group in stance, swing phases, and support force while using a walker.

Variable	Step/Berg	M (SD)	SS	df	MS	F	Effect Size	Post hoc
**Two steps**								
Stance phases	1	2.92 (1.05)	1.04	1	1.04	0.83	0.01	-
	2	3.11 (1.18)						
Swing phases	1	2.01 (0.84)	0.00	1	0.00	0.00	0.00	-
	2	2.01 (0.78)						
Support force	1	137.01 (68.49)	937.44	1	937.44	0.20	0.00	-
	2	142.60 (68.97)						
**Berg group**								
Stance phases	H	3.54 (0.51)	100.60	2	50.30	123.78*	0.68	H, M > L
	M	3.73 (0.81)						
	L	2.08 (0.73)						
Swing phases	H	1.28 (0.42)	37.94	2	18.97	54.41*	0.48	H < M > L
	M	2.64 (0.60)						
	L	2.08 (0.72)						
Support force	H	189.14 (67.89)	159,826.81	2	79913.40	23.46*	0.29	H > M > L
	M	127.62 (43.96)						
	L	101.34 (61.41)						

H: high fall risk, M: medium fall risk, L: low fall risk, * *p* < 0.01.

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
