# Peer review of "Use of Reaction Force to Evaluate Older Adults’ Gait Patterns While Using a Walker to Walk"

_geriatrics, 2019, doi:10.3390/geriatrics4030043_

Round 1

Reviewer 1 Report

This paper presents a method to conduct gait analysis for people who walk with the assistance of a walker without using sensors placed on the person’s body. The analysis presented is interesting, but I think that were the paper framed as a methodological contribution it would strengthen its merit. The conclusion that the study presents regarding those in the medium risk group needs to be thought through in a broader context. While it might be the case that these people’s risk of falling is not fully comprehended limiting these people’s mobility in the pretense of protecting them may lead to a dire outcome as well. The paper needs to be edited so that readers may establish a more coherent understanding of the experiment. Terms and abbreviations need to be introduced before they can be used for example, BBS is used in the abstract without being introduced. Many sentences need to be restructured so that they are logically sound in the English language. Finally I would like to encourage the author to relate to older adults in appropriate language and remove the term ‘elderly’ from the paper.

Reviewer 2 Report

Summary

The paper describes an walker impact evaluation focused on elderly people. The author uses a combined technique of BBS and  GC to evaluate groups of walker users, while this group consists of 64 individuals, they were divided in 3 sub groups of 20 individuals, according with their BBS score.      

Broad comments

- The paper is hard to understand as it main idea is not clearly presented. 

- The author justify the method by mentioning that GC measures in a specialized equipment may be not available, but do not compare the results from the proposed method with it and neither mention how this method may be more available than the first one, as it also uses measurement instruments (line 74)

- Introduction lacks information about the importance of understanding the differences in their stance, swing phases and support force and their relationship with this new method. If the only difference from this method from another is the presence of the equipment, it should be analysed if the results are as precise as using another equipment.

- The same from the last item applies for at the hypothesis. It is not clear in the text why it is important to  prove these 3 hypotheses.

- It is not clear if having 60 volunteers was planned or not, neither are clear the criteria that were utilized to determine this number.

- The experimental setup is described but does not contain any explanation of how it was established.

- Was the study submitted to an ethical committee beforehand? If not and the journal should evaluate, more documents should be sent along with the paper as a extra material (participant consent model, signatures or any other relevant document).  

- Results section needs more explanation and could be better organized. 

- Paper needs more details in the discussion part. What is the impact of these results? Having a bigger time in swing phase (medium group) has what kind of impact? Please detail more the numbers of H, M and L groups for a direct comparison.

- At the current state, it is not clear if conclusions are supported by the results and neither the scientific contribution of the paper. These should be very explicit. It should contain detalis of the main gains of this method/analysis.

- Please do an english revision of the paper before submitting.

- Most of the references are outdated.

- The paper has scientific potential and can be improved with more information about its methodology, results and a better   

Specific comments

Line 12-14: "walker user were elder or disability, the drawback of these measures are obtained in specialized laboratory equipment, which may not be available in daily activities and clinics. 64 elderly volunteers were enrolled." 

- This part of the text is confusing, please be more clear about what is the subject of the study. The first part state the problem, but there is no follow up about the hypothesis to solve it.

Lines 66-70: This paragraph is very confusing. In a way, may induce in thinking that the actual drawbacks are obtained by the equipment.. and I believe it was meant to be that the drawback is the process of obtaining these information with a specialized equipment.  The same consideration applies in the end of the paragraph.

Round 2

Reviewer 1 Report

The authors changed the use of elders to seniors another term that is not considered appropriate in the discussion of older adults. I suggest that they either refrain from using a descriptive term or use 'older adults'

The English language still has problems in the new text. see attached pdf

I would still like to encourage including a discussion regarding limiting the activity of those who are at medium risk.

as this is currently framed as a methodological contribution I suggest adding literature and discussion regarding research in natural settings vs. lab research.

Author Response

Dear Reviewer:

I am very grateful to your suggestions, I had re-word research articles according to your suggestions. According to your opinions, I had list the answer as below, I wish it could helpful for your review:

The authors changed the use of elders to seniors another term that is not considered appropriate in the discussion of older adults. I suggest that they either refrain from using a descriptive term or use 'older adults'

n   Thank you for your suggestion, I had replaced " seniors " with " older adults ".

The English language still has problems in the new text. see attached pdf

n   The article has been revised and the English has been edited again (English-10888).

I would still like to encourage including a discussion regarding limiting the activity of those who are at medium risk.

n   I had reword the last paragraph (please refer to p.6).

as this is currently framed as a methodological contribution I suggest adding literature and discussion regarding research in natural settings vs. lab research.

n   I had reword the section literature and discussion according to your suggestion (please refer to p.2 and p.6).

Reviewer 2 Report

I'm glad that you accepted the suggestions and did a great work in improving the paper. The paper now is easier to read and I believe there is enough information to proceed.

I do not have any major suggestions for the initial sections of the paper.  I do have, however, for the discussion section.

My suggestions:

- If you can (if they exist) use references from the last 5 years,10 years top. This was improved in comparison with the first version, but I believe it can still be improved. (but it is not a major problem and you can overlook it)

- Line 206, 207:

"The evaluation of gait in this study can indeed reduce the requirement to wear a large number of instruments, " - I suggest stopping the paragraph idea here and explain more about this conclusion.

- Line 207, 208:

"Seniors can be initially evaluated by a caregiver at a nursing center/rehabilitation center" - Explore more this idea, what this means? Is it good? bad? What or how does this impacts today daily routines?   

- Line 208-2010:

", but daily walking includes up and down-hill, up and down stairs, and even ground materials and other factors. For stability when walking, future research is obviously required; however, this is an exciting first step." - Start this idea in another paragraph. It delimits your study, and you may connect with the following idea of future work. Please, describe at least one or two possible future work. What are the next steps? What can be researched  using your paper as reference?

Author Response

Dear Reviewer:

I am very grateful to your suggestions, I had re-word research articles according to your suggestions. According to your opinions, I had list the answer as below, I wish it could helpful for your review:

-Line 206, 207:

"The evaluation of gait in this study can indeed reduce the requirement to wear a large number of instruments, " - I suggest stopping the paragraph idea here and explain more about this conclusion.

n   Thank you for your suggestion, I had reword the last paragraph including the research findings and future applications. (please refer to p.6).

- Line 207, 208:

"Seniors can be initially evaluated by a caregiver at a nursing center/rehabilitation center" - Explore more this idea, what this means? Is it good? bad? What or how does this impacts today daily routines?   

n   Thank you, I had reword the this paragraph including the research findings and future applications, please refer to p.6.

- Line 208-2010:

", but daily walking includes up and down-hill, up and down stairs, and even ground materials and other factors. For stability when walking, future research is obviously required; however, this is an exciting first step." - Start this idea in another paragraph. It delimits your study, and you may connect with the following idea of future work. Please, describe at least one or two possible future work. What are the next steps? What can be researched using your paper as reference?

n   Thank you, I rewritten and organized this paragraph, please refer to p.6.